# Identification of the Actinomycin D Biosynthetic Pathway from Marine-Derived *Streptomyces costaricanus* SCSIO ZS0073

**DOI:** 10.3390/md17040240

**Published:** 2019-04-23

**Authors:** Mengchan Liu, Yanxi Jia, Yunchang Xie, Chunyan Zhang, Junying Ma, Changli Sun, Jianhua Ju

**Affiliations:** 1CAS Key Laboratory of Tropical Marine Bio-resources and Ecology, Guangdong Key Laboratory of Marine Materia Medica, South China Sea Institute of Oceanology, Chinese Academy of Sciences, 164 West Xingang Road, Guangzhou 510301, China; 18696164503@163.com (M.L.); jiayanxi0928@163.com (Y.J.); xieyunchang1984@sina.com (Y.X.); zhchuny@foxmail.com (C.Z.); majunying@scsio.ac.cn (J.M.); lingluboxi@163.com (C.S.); 2College of Oceanography, University of Chinese Academy of Sciences, Beijing 100049, China

**Keywords:** actinomycin D, biosynthetic gene cluster, biosynthesis, bioproducer, *Streptomyces costaricanus*

## Abstract

Bioactive secondary metabolites from *Streptomycetes* are important sources of lead compounds in current drug development. *Streptomyces costaricanus* SCSIO ZS0073, a mangrove-derived actinomycete, produces actinomycin D, a clinically used therapeutic for Wilm’s tumor of the kidney, trophoblastic tumors and rhabdomyosarcoma. In this work, we identified the actinomycin biosynthetic gene cluster (BGC) *acn* by detailed analyses of the *S. costaricanus* SCSIO ZS0073 genome. This organism produces actinomycin D with a titer of ~69.8 μg mL^−1^ along with traces of actinomycin X*_oβ_*. The *acn* cluster localized to a 39.8 kb length region consisting of 25 open reading frames (ORFs), including a set of four genes that drive the construction of the 4-methyl-3-hydroxy-anthranilic acid (4-MHA) precursor and three non-ribosomal peptide synthetases (NRPSs) that generate the 4-MHA pentapeptide semi-lactone, which, upon dimerization, affords final actinomycin D. Furthermore, the *acn* cluster contains four positive regulatory genes *acnWU4RO*, which were identified by in vivo gene inactivation studies. Our data provide insights into the genetic characteristics of this new mangrove-derived actinomycin D bioproducer, enabling future metabolic engineering campaigns to improve both titers and the structural diversities possible for actinomycin D and related analogues.

## 1. Introduction

Actinomycins are a group of chromopeptide lactone antibiotics. To date, 42 actinomycins have been isolated and identified from many species of *Streptomyces* (Appendix A), including actinomycin D, *N*-demethylactinomycins, actinomycin C, actinomycin F, actinomycin Z, actinomycin G and actinomycin Y [1,2,3] (Appendix A). Some actinomycin analogues, such as methylated actinomycin D [4], an actinomycin Z analogue having an additional oxygen bridge between the chromophore and β-depsipentapeptide [5], actinomycins D1–D4 [6] and neo-actinomycins A and B [7], possess structurally modified cyclopeptide rings or chromophore (Appendix A). Initiatives to develop actinomycin analogues with superior bioactivities have been heavily rooted in precursor-directed biosynthesis [8]. Actinomycins, commonly employed clinically in anticancer therapeutic regimes, exhibit excellent antitumor activity. The core phenoxazinone chromophore intercalates between the stacked nucleobases at guanine/cytosine sites of DNA whereas the pentapeptide elements bind to the minor groove; these binding interactions effectively inhibit duplication and transcription processes in tumor cells [9]. Exemplary in this fashion, actinomycin D is a highly effective chemotherapeutic used to treat Wilm’s kidney tumors, trophoblastic tumors and rhabdomyosarcoma [2]. Actinomycin D also specifically targets and down-regulates the expression of stem-cell transcription factor, Sox-2; this protein facilitates depletion of the stem-cell population resulting in the inability of breast cancer cells to initiate tumor progression [10]. Low doses of actinomycin D specifically activate p53-dependant transcription enhancing the activity of chemotherapeutic drug-induced killing of p53 positive human tumor cells [11]. Actinomycins are also able to inhibit some viruses such as the coxsackie virus B3 and HIV-1, the causative agent of AIDS [12,13]. The combination of these novel activities and potential in human health makes actinomycin D one of the most promising candidates for medicinal development campaigns. Moreover, the actinomycins often display strong antimicrobial activities. Potent antibacterial activities against human pathogens, human pathogenic fungi, as well as aquatic pathogenic bacteria are well known [14]. For instance, actinomycin X_2_ kills both resting and budding spores of *Bacillus megaterium* and actinomycin X*_oβ_* has pronounced anti-tuberculosis activities [15]. Notably, despite these very favorable activities, clinical applications of actinomycin D are often limited by undesirable side-effects; liver and kidney damage rank high on this list of detrimental side-effects [16].

Recent progress in genome research has revealed the presence of the actinomycin D biosynthetic gene cluster (BGC) in *S. chrysomallus*. The gene cluster was found to encompass 50 kb of contiguous DNA on the chromosome of *S. chrysomallus* and to contain 28 biosynthetic genes bordered on both sides by IS (Insertion Sequence) elements [17]. Remarkably, 9 of the genes embedded within the actinomycin D BGC have two copies, all of which are in the same order but in the opposite orientation; such an arrangement of genes within a BGC is unprecedented. Sequencing of the actinomycin BGC in *S. antibioticus* IMRU 3720, which produces actinomycin D, also revealed 20 genes organized into a similar framework but without gene duplicates as in the actinomycin D biosynthetic gene cluster of *S. chrysomallus* [18]. In addition, the actinomycin G BGC in *S. iakyrus* was identified and reported in 2013 [19].

A defining feature of the actinomycins is a central phenoxazinone chromophore which serves to bridge two pentapeptide lactones consisting of diverse amino acids. The pentapeptide precursor is biosynthesized by the non-ribosomal peptide synthetase (NRPS) assembly line with the 4-methyl-3-hydroxy-anthranilic acid (4-MHA) as the initiating unit [20,21]. The biosynthesis of 4-MHA differs in various actinomycin-producing strains. For example, in *S. chrysomallus*, *Streptosporangium sibiricum* and *S. parvulus*, tryptophan is implicated as a substrate for tryptophan-2,3-dioxygenase which primarily forms the important MHA precursor, *N*-formyl kynurenine (Figure 2). *N*-formyl kynurenine then serves as a substrate for kynurenine formamidase and kynurenine 3-monooxygenase; the actions of these two enzymes produce 3-hydroxykynurenine (3-HK) and this sequence of chemistries is apparent in all actinomycin producers. Methylation of 3-HK then affords 4-methyl-3-hydroxy-kynurenine (4-MHK). 4-MHK is then catalyzed by hydroxykynureninase to form 4-MHA. Some deviations in this chemistry are seen, however, based on the precise actinomycin producer in question. For instance, in *S. antibioticus*, tryptophan is converted to 3-hydroxyanthranilic acid (3-HA) which is subsequently processed by a methyltransferase to form 4-MHA [22].

Perhaps one of the more intriguing questions associated with actinomycin assembly has to do with the final step of actinomycin D biosynthesis and how two molecules of MHA-pentapeptide converge, via phenoxazinone assembly, to generate the intact actinomycin. Phenoxazinone synthase (PHS), a 650 aa, two copper-containing phenoloxidase, has long been theorized to mediate this final step [23]. Once cloned and expressed in vitro, PHS was characterized and anticipated to activate MHA-pentapeptide monomers for subsequent dimerization reactions [24,25]. However, in the actinomycin-producer *S*. *antibioticus* IMRU 3720, actinomycin D production persisted in an inactivated ∆*phsA* mutant [26], suggesting that the putative phenoxazinone synthase is not essential for actinomycin assembly. Hence, the mechanism by which the two MHA-pentapeptide monomers come together to form intact actinomycins remains largely unclear.

Enabled by high quality genomic scanning and analyses, we report here the identification of a distinct 39.8 kb gene cluster from *S. costaricanus* SCSIO ZS0073, a previously identified actinomycin D and actinomycin X*_oβ_* producer [27]. We have defined the gene cluster boundaries and described the organization of the complete biosynthetic gene cluster as guided by the results of gene insertions and metabolic profile analyses. Interestingly, we have discovered that, within the upstream and downstream regions of the cluster, can be found regulatory genes *acnW, acnR, acnU4* and *acnO*. Our study expands insights into how actinomycin titers and structural diversities stand to be improved via the application of combinatorial biosynthetic approaches. Holistically speaking, actinomycin analogs with reduced toxicities and improved bioactivities represent important goals for the downstream application of new knowledge reported herein.

## 2. Results

### 2.1. Identification of the Actinomycin D Biosynthetic Gene Cluster

To identify the gene cluster, we sequenced the *S. costaricanus* SCSIO ZS0073 genomic DNA using a combination of Hiseq 4000 and PacBio smart technologies [27]. *S. costaricanus* SCSIO ZS0073 is a mangrove-derived *actinomycete* which produces antibacterial secondary metabolites such as fungichromin, actinomycin D and actinomycin X*_oβ_* [27,28]. We analyzed a 39.8 kb region containing 24 open reading frames (ORFs), suspected of coding for actinomycin D biosynthesis. We elucidated the gene organization as shown in Figure 1 and deduced its biosynthetic pathway (Figure 2) along with assigned gene product functions as noted below in Table 1. The gene cluster has been deposited in Genbank (with the accession number of MK234849). A cosmid library of *S. costaricanus* SCSIO ZS0073 was constructed using the SuperCos 1 vector system and 2 positive clones (9C2, 9A7) covering the whole gene cluster were screened using targeted gene inactivations.

### 2.2. Determination of the Actinomycin D Biosynthetic Gene Cluster Boundaries

The upstream boundary of the actinomycin D gene cluster was preliminarily determined as being between *orf*(−1) and *acnW*; the downstream boundary was defined between *acnT3* and *orf*(+1). The *orf*(−1) encodes Type VII secretion system protein EccC. Although *orf*(−1) carries with it the function of enabling secretion, all the inactivation mutants of *orf*(−1) failed to show any differences in actinomycin D production levels compared to wild-type (WT) (Appendix A). The upstream boundary region was further delineated by a set of gene inactivations for *orf*(−1), *orf*(−2) and *orf*(−3). None of the inactivation bearing mutants for these genes displayed discernible differences in actinomycin D biosynthesis rates or yields relative to WT producer (Appendix A). Likewise, the downstream boundary was ascertained by bioinformatic analyses and the application of mutants bearing inactivated *orf*(+1), *orf*(+2), *orf*(+3) genes. Although *orf*(+1) gene encodes an *N*-acetyltransferase, it is shown by inactivation experiments, the mutant strains produce the same level of actinomycin D compared to wild type strains. Further evaluations proximal to the downstream boundary revealed that genes corresponding to *orf*(+2) and *orf*(+3) encode a hypothetical protein and a two-component hybrid sensor and regulator; comparative analyses of ∆*orf*(+2) and ∆*orf*(+3) inactivation mutants (vs. WT) revealed that neither of these genes and their respective products play a role in actinomycin D biosynthesis (Appendix A).

### 2.3. NRPS Genes for Peptide Chain Assembly in Actinomycin D Biosynthesis

Following the biosynthesis of 4-MHA, this anthranilic acid derivative, along with other amino acids, by several NRPSs ultimately affording the 4-MHA-pentapeptide halves of actinomycin (Figure 2A). The pentapeptide is composed of five different amino acids: l-threonine, d-valine, l-proline, l-sarcosine and methyl-valine. The genes responsible for peptide chain assembly in *S. costaricanus* SCSIO ZS0073 are *acnD*, *acnE*, *acnN1*, *acnN2*, *acnN3*. AcnD was found to contain 66 amino acid residues arranged in a manner that resembles the MbtH protein. MbtH-like proteins are found in many NRPS gene clusters [29] and researchers have demonstrated that MbtH-like proteins participate in some adenylation reactions by tightly binding to NRPS proteins containing adenylation domains [30,31]. MbtH-like proteins, such as gene *dptG* in *S. roseosporus* participate in the biosynthesis of the cyclic lipotridecapeptide antibiotic daptomycin [32] and *mbtH* in *Mycobacerium* spp. responsible for the siderophore biosynthesis [33]. The NRPS gene *acnN1* encodes an adenylation domain protein and *acnE* encoding 4-MHA carrier protein. Two large multidomain NRPSs *acnN1* and *acnN3* are responsible for peptide chain assembly and release from the NRPS machinery. Gene deletions of *acnD* + *acnE* in combination, *acnN1* and *acnN3* were found to abolish actinomycin D biosynthesis (Figure 3) in the mutant producers bearing these gene inactivations.

### 2.4. Biosynthetic Genes of 4-MHA

We identified a set of genes, designated as *acnG*, *acnH*, *acnL* and *acnM*, that show high functional similarities to enzymes involved in the biosynthesis of 4-MHA, the critical building block of the actinomycin chromophore. These genes are homologous to a set of four counterparts in actinomycin-producing strains *S. antibioticus* IMRU3720 and *S. chrysomallus* [17,18]. These genes encode for arylformamidase (*acnG*), tryptophan-2, 3-dioxygenase (*acnH*), kynureninase (*acnL*) and methyltransferase (*acnM*). The gene *acnM* shows 88% identities with the 3-hydroxy kynurenine methyltransferase acmL, which is responsible for the methylation of 3-HK [34]. However, the vital step entailing kynurenine conversion to 3-hydroxykynurenine appears to be catalyzed by a kynurenine 3-monooxygenase, the gene for which is not found within the *acn* gene cluster. Inactivation of *acnGHLM* in combination, followed by metabolite analysis of the mutant strain revealed that these genes play essential roles in actinomycin D biosynthesis (Figure 3).

### 2.5. Regulatory and Self-Resistance Genes

We identified six genes *acnW*, *acnU4*, *acnR*, *acnT1*, *acnT2* and *acnT3* with apparent regulatory and/or protective functions. *AcnR* encodes a TetR family transcriptional regulator and usually acts as a transcriptional repressor or activator in many biological processes such as cell-cell communication and metabolite regulation [35,36]. TetR family transcriptional regulators belong to a one-component system and generally possess a two-domain structure composed of an N-terminal HTH DNA-binding motif and a C-terminal ligand regulatory domain. Many of these regulators control the expression of transporters, which help bacteria acclimate to their environment [37,38]. In some cases, members of the TetR family suppress the biosynthesis of antibiotics. For instance, disruption of *calR3*, a TetR family member in the calcimycin BGC, was found to improve calcimycin titers [39]. Moreover, *TrdK,* which shows high similarity to TetR family members and is involved in tirandamycin production from marine-derived *Streptomyces* sp. SCSIO 1666 was found by gene inactivations, to suppress tirandamycin biosynthesis; *trdK* inactivation led to dramatically improved tirandamycin titers [40]. The inactivation of *acnR* completely abolished actinomycin D production, consistent with this gene’s essential role as a and important positive regulator of actinomycin D biosynthesis (Figure 3, trace viii). Gene *acnW* is located downstream of the boundary gene *orf(−1)* and appears to encode for a hypothetical protein; *acnW* disruption impaired, but did not abolish, production of actinomycin D. The gene *acnU4* also encodes a hypothetical protein and, as with *acnW*, inactivation of *acnU4* correlates to diminished actinomycin D production (Figure 3). On the basis of these findings, AcnW and AcnU4 appear to function as positive regulators of actinomcyin D production although the structural families to which they belong are not yet apparent. *AcnO* encodes a LmbU-like protein. *LmbU* is a regulatory gene involved in licomycin biosynthesis in *S. lincolnensis* 78-11. It always contains a TTA codon close to the N-terminal end of its ORF. The codon is often found in genes involved in the regulation of differentiation or secondary metabolism [41], in neither case is a specific role in actinomycin D assembly known, *S. costaricanus* SCSIO ZS0073 inactivation mutant strain of ∆*acnO* was not able to produce even trace amounts of actinomycin D (Figure 3, traces ix). Thus, *acnO* might serve as a positive regulator in actinomycin D biosynthetic pathway.

Beyond these gene–function correlations, we also identified three putative transporter genes *acnT1*, *acnT2* and *acnT3*. The *acnT1* gene is predicted to encode an ATP-binding cassette (ABC) transporter which uses the energy of ATP to transport molecules through the membrane. Bioinformatics analyses suggest that *acnT2* encodes a multidrug ABC transporter permease whereas *acnT3* encodes a UvrA-like protein, which is a DNA binding protein involved in the excision repair of DNA. Somewhat surprisingly, inactivations of *acnT1 acnT2* and *acnT3* all produced mutant strains able to generate actinomycin D with efficiencies rivaling those seen with the WT producer (Figure 3). Consequently, we concluded that these three transporter genes are not necessary in actinomycin D biosynthesis.

### 2.6. Nonessential Genes of Unknown within the acn Cluster

The BGC for actinomycin D in *S. costaricanus* SCSIO ZS0073 houses a number of genes not associated with actinomycin D assembly. For instance, close to the upstream boundary reside genes *acnA* and *acnB;* all of these encode for hypothetical proteins. Additionally, *acnU1*, *acnU2*, *acnU3* also encode for proteins whose functions have not been deciphered and whose roles in actinomycin D construction appear nonessential. Inactivations for all these genes have no impact on actinomycin D biosynthesis relative to WT production efficiencies (Figure 4). Bioinformatics have revealed that a*cnC* resides between *acnU2* and *acnU3* and codes for an acyl-CoA dehydrogenase. This gene’s inactivation also appears to have no impact upon actinomycin D synthesis compared to the WT strain. This too is the case for *acnQ*. Located downstream of the *acnR*, *acnQ* was proposed, based on bioinformatics, to code for a siderophore-interacting protein. Such small molecule siderophore-interacting systems often constitute transport systems by which microbes control intracellular concentrations of specific secondary metabolites. As with the other genes noted however, inactivation of *acnQ* was found to have no bearing whatsoever on actinomycin D production compared to the WT producer (Figure 4).

### 2.7. Cytochrome P450 Gene acnP Is Responsible for the Hydroxylation of Proline in Actinomycin X_oβ_

X-type actinomycins contain 4-hydroxyproline (actinomycin X*_oβ_*) or 4-oxoproline (actinomycin X_2_) in their β-pentapeptide lactone rings whereas their α-ring contains proline. Previous studies demonstrated the importance of a 4-oxoproline synthase within the actinomycin BGC of *S. antibioticus*. This enzyme catalyzes proline oxidation in each of the actinomycin halves (prior to dimerization) to form 4-hydroxyproline or 4-oxoproline; condensation of each of the hydroxylated actinomycin halves affords actinomycin X*_oβ_* or actinomycin X_2_ [42]. Within the actinomycin BGC in *S. costaricanus* SCSIO ZS0073, we identified a cytochrome P450 gene, *acnP*, that encodes a 436 aa protein. We hypothesized that AcnP is responsible for proline hydroxylation *en route* to actinomycin X*_oβ_*. This speculation was validated by targeted inactivation of *acnP*; fermentations and metabolite analyses of the ∆*acnP* mutant revealed the complete abrogation of actinomycin X*_oβ_* production and concomitant production of actinomycin D with titers rivaling those seen with the WT producer (Figure 5). These data make clear that AcnP is responsible for the 4-oxoproline found in actinomycin X*_oβ_*.

The PHS gene outside the *acn* cluster is not necessary for actinomycin D biosynthesis. PHS has been presumed to catalyze the oxidative condensation of two 4-MHA-pentapeptide lactone “monomers” in the last step of actinomycin D biosynthesis. Previous studies had, in fact, demonstrated that PHS is involved in a variety of enzymatic condensations of *ortho*-aminophenols to form phenoxazinones. Within the genome of *S. costaricanus* SCSIO ZS0073, we identified a *phs orf* encoding a 627 aa, two copper-containing phenoloxidase separated from the actinomycin biosynthetic cluster by 5.77 Mbp. This *phs* gene shows sequence similarity (84% identity) to *phsA* in *S. antibioticus* IMRU3720. To validate this gene’s involvement in actinomycin D assembly, we generated a ∆*phs* mutant strain and assessed its biosynthetic capacity relative to the WT producer. Surprisingly, *phs* inactivation in the *S. costaricanus* SCSIO ZS0073 strain had no detectable impact upon actinomycin D production relative to the WT producer (Figure 4). This result is consistent with the previous gene inactivation result for *phsA* in *S. antibioticus* IMRU3720 [26] and indicated also, that for *S. costaricanus* SCSIO ZS0073 actinomycin D assembly does not appear to require a PHS.

*acnF* is essential to actinomycin D biosynthesis. Within the actinomycin cluster, *acnF* encodes for a hypothetical protein with 212 aa. Gene inactivation for *acnF*, revealed that *acnF* is indispensable for actinomycin D construction (Figure 4). *S. costaricanus* SCSIO ZS0073 inactivation mutant strain of ∆*acnF* was not able to produce even trace amounts of actinomycin D (Figure 4, traces x), thus showcasing the importance of its gene products to actinomycin D biosynthesis. We proposed *acnF* might be involved in the dimerization of the 4-MHA-pentapeptide monomer *en route* to actinomycin (Figure 2). Efforts to identify the exact roles played by *acnF* products are currently ongoing and will be reported in due course.

## 3. Materials and Methods

### 3.1. General Experimental Procedures

Reagents for polymerase chain reactions (PCR) were purchased from Takara Co. (Dalian, China) and Trans Gene Co. (Beijing, China). The plasmid kit and gel extraction kit were from Promega. Unless otherwise indicated, other biochemicals and chemicals were purchased from standard commercial sources and used without further purification. All DNA manipulations were conducted according to the standard procedures or manufactures’ instruction. DNA and aa sequence analyses were performed with the seqmen and editsequence in the Lasergene software package (DNASTAR, Madison, WI, USA). All primers and reagents used in this work were purchased from Sangong Bio-Pharm Technology Co., Ltd., Shanghai, China. 

### 3.2. Bacterial Strains, Plasmids and Culture Conditions

*S. costaricanus* SCSIO ZS0073, a marine actinomycete isolated from the red sand park of Guangxi (China), was used to identify the actinomycin D gene cluster [27]. The strain and mutants were grown at 30 °C on ISP-2 medium (with 3% sea salt added) for cultivation. *E. coli* was used for DNA cloning and sequencing. *E. coli* ET12567/pUZ8002, which is methylation deficient, was employed as the donor cell for conjugal transfer of DNA into *S. costaricanus* SCSIO ZS0073. All *E. coli* strains were grown in liquid lysogeny broth (LB) at 37 °C or 30 °C and 200 rpm**.** When used, antibiotics were added at the following concentrations: chloramphenicol (Chl, 25 μg mL^−1^), apramycin (Apr, 50 μg mL^−1^), kanamycin (Kan, 50 μg mL^−1^), ampicillin (Amp, 100 μg mL^−1^). The plasmids SuperCos 1 and pIJ773 were used for *S. costaricanus* SCSIO ZS0073 genomic library construction and the *aac(3)IV-oriT* resistance gene amplifying, respectively.

### 3.3. Whole Genome Scanning and Sequence Analysis

*S. costaricanus* SCSIO ZS0073 genomic DNA was isolated according to the protocol with slight modification [43]**.** Whole-genome scanning was achieved using both 454 and Solexa technology at Macrogen (Seoul, Korea). Assembly, annotation and bioinformatics analyses allowed us to define the correct contiguous fragments corresponding to the acn cluster. Assignments of ORFs and their functional predictions were accomplished using FramePlot 4.0 (http://nocardia.bih.go.jp/fp4) and Blast (http://blast.ncbi.nlm.nih.gov/) software packages.

### 3.4. Genomic Library Construction and Screening

Genomic DNA was partially digested with Sau3AI and 30–40 kb fragments were ligated into XbaI/BamHI digested and dephosphorylated SuperCos 1. The resulting ligation mixture was packaged with Gigapack III gold and transduced into *E. coli* LE392 to generate the genomic library, according to the manufacturer instructions.

### 3.5. Inactivation of S. costaricanus SCSIO ZS0073 by λ-RED-Mediated PCR-Targeting Mutagenesis

Targeted genes in the *S. costaricanus* SCSIO ZS0073 biosynthetic gene cluster were inactivated using λ-mediated PCR-targeting methodology [44]. Two cosmids were used to disrupt target genes. An apramycin resistance cassette *aac(3)IV-oriT* fragment obtained by PCR (digested pIJ773 was used as template), with primer pairs containing 39-nucleotide extensions derived from the 5′- and 3′-ends of the targeted genes, was used to replace an internal region of each of the targeted genes. Each of the mutated genes was verified by PCR with primers designed to be 10–300 bp outside of the disruption region, with verification by restriction enzyme digestion. The constructed mutated cosmids were introduced into non-methylating *E. coli* ET12567/pUZ8002 for conjugal transfer. For conjugation, harvested *S. costaricanus* SCSIO ZS0073 spores were suspended in TSB medium and incubated for 3–5 h at 28 °C and 200 rpm after heating for 10 min at 50 °C. The culture was then centrifuged to harvest the germinated spores as the conjugation recipients. At the same time, LB supplied with Kan (50 μg mL^−1^), Chl (25 μg mL^−1^) and Apr (100 μg mL^−1^) was inoculated with *E. coli* ET12567/pUZ8002 containing each mutated cosmid. After the culture OD_600_ increased to 0.6–0.8, the cells were harvested, washed three times with LB, resuspended in 500 μL LB medium and mixed with the previously germinated spores. The mixture was plated on modified ISP-4 medium containing MgCl_2_ (60 mM). The plates ware incubated at 28 °C for 18–20 h, then each plate was covered with sterile deionized water (1 mL), trimethoprim stock solution (Tmp, 30 μL, 50 mg mL^−1^) and Apr stock solution (25 μL, 50 mg mL^−1^). Finally, all plates were incubated at 28 °C for an additional 6–7 days until exconjugants appeared. Double-crossover mutants were primarily selected on the basis of Kan^S^Apr^R^ phenotypes and the desired double crossover mutants were further verified by PCR with primers listed in Appendix A.

### 3.6. Fermentation and Analysis of S. costaricanus SCSIO ZS0073 WT and Mutant Strains

*S. costaricanus* SCSIO ZS0073 WT and mutant strains were inoculated into 250 mL flasks with 50 mL liquid ISP-2 medium and incubated on a rotary shaker at 28 °C, 200 rpm. After 7 days fermentation, each of the 50 mL cultures was added to 100 mL butanone and then vigorously mixed for 30 min. The butanone phase was separated and then evaporated to dryness to yield a residue. The residue was dissolved in 2 mL methanol and centrifuged and the resulting supernatant then subjected to HPLC analysis. Analytical HPLC was performed on an Agilent 1260 HPLC system (Agilent Technologies Inc., USA) equipped with a binary pump and a diode array detector using a Phenomenex Prodigy ODS column (150 × 4.60 mm, 5 μm) with UV detection at 254 nm. The mobile phase was comprised of solvents A and B. Solvent A consisted of 15% CH_3_CN in water supplemented with 0.1% TFA whereas solvent B consisted of 85% CH_3_CN in water supplemented with 0.1% TFA. Samples were eluted with a linear gradient from 5% to 90% solvent B in 20 min, followed by 9–100% solvent B for 5 min, then eluted with 100% solvent B for 3 min, at a flow rate of 1 mL/min and UV detection at 254 nm. LC-ESI-MS data were obtained using an amaZon SL ion trap mass spectrometer (Bruker, Billerica, MA, USA). The mobile phase comprises solvents A and B. Solvent A consists of 100% ddH_2_O supplemented with 0.1% methanoic acid whereas solvent B consists of 100% CH_3_CN supplemented with 0.1% methanoic acid. Samples were eluted with a linear gradient from 5 % to 90% solvent B in 20 min, followed by 9–100% solvent B for 5 min, then eluted with 100% solvent B for 3 min, at a flow rate of 1 mL/min and UV detection at 254 nm.

## 4. Conclusions

Actinomycin D is a vital antibiotic in the treatment of Wilms’ tumor, trophoblastic tumors and rhabdonyosarcoma. Here, we have identified and characterized a new actinomycin D BGC (*acn*) in marine derived producer *S. costaricanus* SCSIO ZS0073 by carrying out whole genome sequencing and systematic gene disruptions. Relative to the gene cluster for actinomycin of *S. chrysomallus* and *S. antibioticus* IMRU3720, the size of the *acn* cluster in *S. costaricanus* SCSIO ZS0073 is smaller. In silico analysis, gene inactivation and metabolomics data enabled us to deduce the biosynthetic pathway leading to actinomycin D in *S. costaricanus* SCSIO ZS0073. We furthermore determined the *acn* cluster boundaries and elucidated the functions of positive regulatory genes *acnWU4RO* along with the cytochrome P450 gene *acnP* which is responsible for installing the 4-oxoproline seen in actinomycin X*_o__β_*. The discovery of this new actinomycin BGC advances initiatives to engineer new actinomycin D analogues for clinical use and to explore the still elusive mechanism of actinomycin 4-MHA-pentapeptide monomer dimerization *en route* to intact actinomycins.

## Figures and Tables

**Figure 1 marinedrugs-17-00240-f001:**
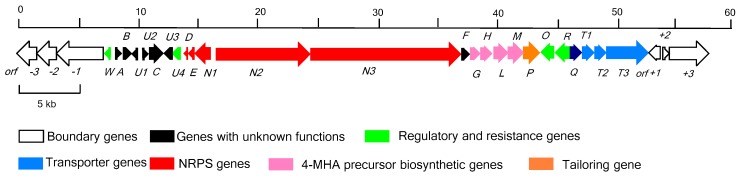
Gene organization of the actinomycin D cluster in *S. costaricanus* SCSIO ZS0073. The direction of transcription and the proposed functions of individual ORF are indicated.

**Figure 2 marinedrugs-17-00240-f002:**
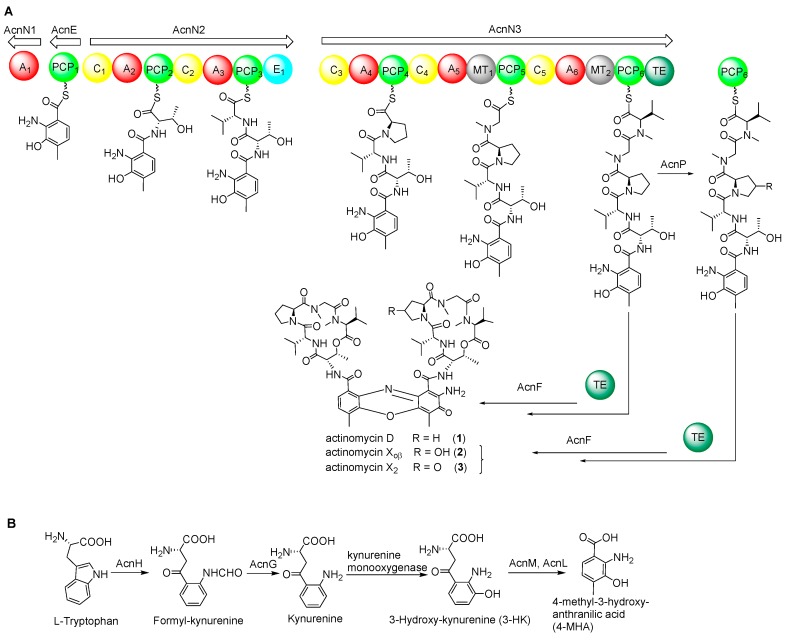
Proposed model for actinomycin D assembly (**A**) and 4-MHA precursor production (**B**) in *S. costaricanus* SCSIO ZS0073. A: adenylation domain; C: condensation; PCP: peptidyl carrier protein; TE: thioesterase.

**Figure 3 marinedrugs-17-00240-f003:**
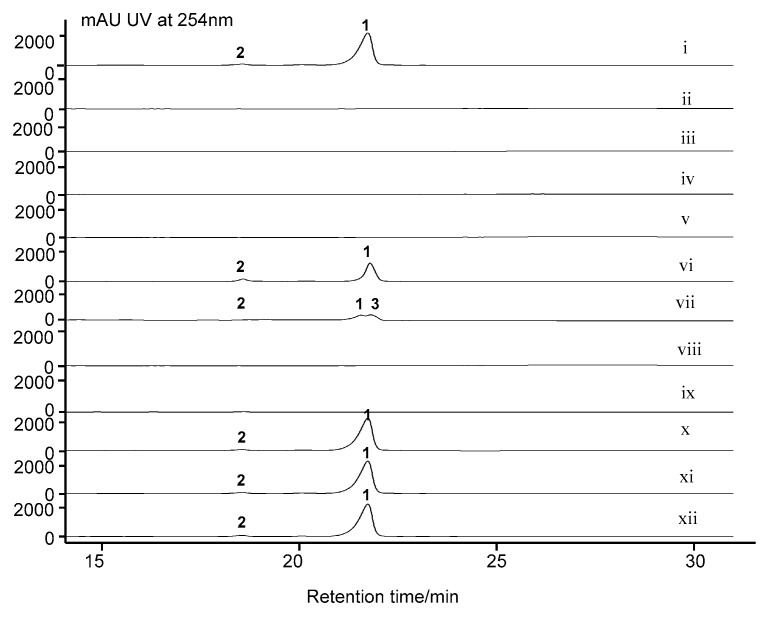
HPLC analysis of *S. costaricanus* strains fermentation extracts. (i) WT producer; (ii) ∆*acnDE* mutant; (iii) ∆*acnN1* mutant; (iv) ∆*acnN3* mutant; (v) ∆*acnGHLM* mutant; (vi) ∆*acnW* mutant; (vii) ∆*acnU4* mutant; (viii) ∆*acnR* mutant; (ix) ∆*acnO* mutant; (x) ∆*acnT1* mutant; (xi) ∆*acnT2* mutant; (xii) ∆*acnT3* mutant. 1, actinomycin D; 2, actinomycin X*_oβ_*.; 3, actinomycin X_2._

**Figure 4 marinedrugs-17-00240-f004:**
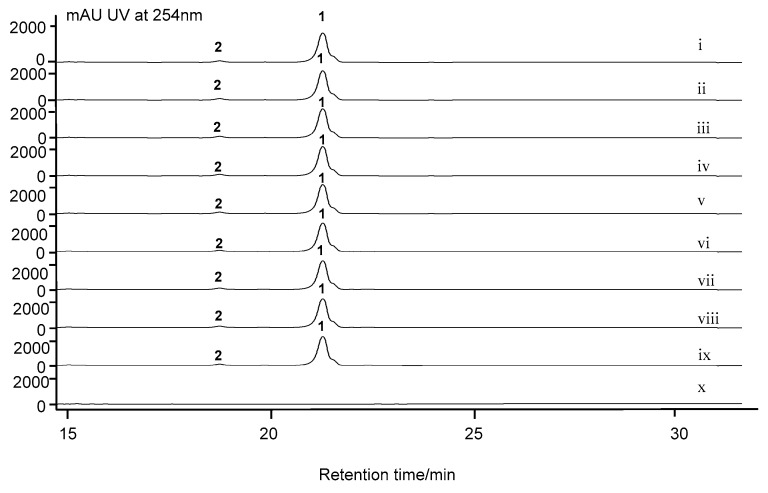
HPLC analyses of *S. costaricanus* strains fermentation extracts. (i) WT producer; (ii) ∆*acnA* mutant; (iii) ∆*acnB* mutant; (iv) ∆*acnC* mutant; (v) ∆*acnU1* mutant; (vi) ∆*acnU2* mutant; (vii) ∆*acnU3* mutant; (viii) ∆*phs* mutant; (ix) ∆*acnQ* mutant; (x) ∆*acnF* mutant. 1, actinomycin D; 2, actinomycin X*_oβ_*.

**Figure 5 marinedrugs-17-00240-f005:**
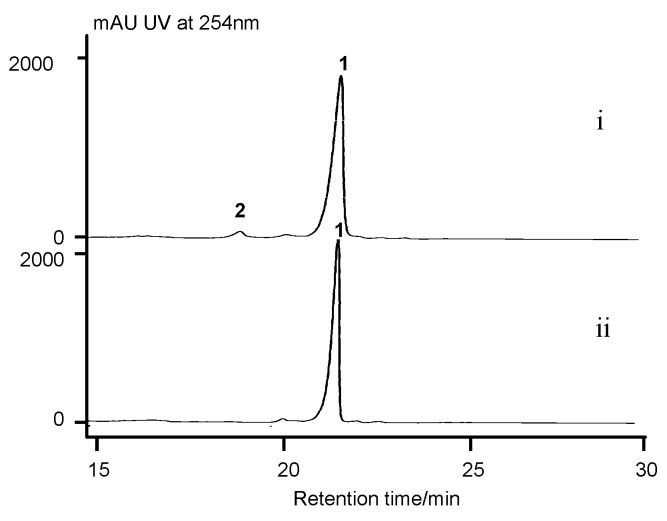
HPLC analysis of *S. costaricanus* strains fermentation extracts. (i) WT producer; (ii) ∆*acnP* mutant. 1, actinomycin D; 2, actinomycin X*_oβ_*.

**Table 1 marinedrugs-17-00240-t001:** Deduced *orf* functions in the *acn* cluster.

Gene	Size ^a^	Protein Homolog and Origin	ID/SM (%)	Origin (Protein ID)
*orf*(−3)	492	type VII secretion protein EccB	96/97	SCF88120 *Streptomyces* sp. LamerLS-31b
*orf*(−2)	464	type VII secretion integral membrane protein EccD	99/99	SCF88128 *Streptomyces* sp. LamerLS-31b
*orf*(−1)	1289	Collagen triple helix repeat-containing protein	53/75	*Clostridium uliginosum*
*acnW*	103	Hypothetical protein	99/100	AB905443.1 *Streptomyces rochei* 7434AN4
*acnA*	121	Protein of unknown function	93/96	SCF88143 *Streptomyces* sp. LamerLS-31b
*acnB*	415	Anti-anti-sigma regulatory factor	96/97	SCF88151 *Streptomyces* sp. LamerLS-31b
*acnU1*	121	hypothetical protein GA0115258_11557	99/100	SCF88158 *Streptomyces* sp. LamerLS-31b
*acnU2*	124	hypothetical protein GA0115258_11558	95/96	SCF88164 *Streptomyces* sp. LamerLS-31b
*acnC*	392	acyl-CoA dehydrogenase	91/95	AKJ14982 *Streptomyces* incarnatus
*acnU3*	207	hypothetical protein	80/91	WP_030987180 *Streptomyces*
*acnU4*	187	hypothetical protein	82/88	WP_030592054 *Streptomyces anulatus*
*acnD*	66	protein mbtH	91/95	OOQ48080 *Streptomyces antibioticus*
*acnE*	78	4-MHA carrier protein	73/85	ADG27356 *Streptomyces anulatus*
*acnN1*	467	Adenylation domain protein	51/65	SCD96508 *Streptomyces* sp. DvalAA-43
*acnN2*	2589	non-ribosomal peptide synthase	73/81	WP_057667184 *Streptomyces anulatus*
*acnN3*	4249	non-ribosomal peptide synthetase	78/85	WP_064726364 *Streptomyces parvulus*
*acnF*	211	hypothetical protein	84/88	OOQ48467 *Streptomyces antibioticus*
*acnG*	326	arylformamidase	79/85	SCF58504 *Streptomyces* sp. Cmuel-A718b
*acnH*	284	tryptophan 2, 3-dioxygenase	84/88	OOQ48467 *Streptomyces antibioticus*
*acnL*	420	kynureninase	86/90	OOQ48077 *Streptomyces antibioticus*
*acnM*	346	methyltransferase	88/91	WP_030594248 *Streptomyces anulatus*
*acnP*	386	cytochrome P450	86/91	WP_030594247 *Streptomyces anulatus*
*acnO*	224	LmbU-like protein	73/82	ADG27350 *Streptomyces anulatus*
*acnR*	282	TetR family transcriptional regulator	78/86	OOQ48074 *Streptomyces antibioticus*
*acnQ*	294	siderophore-interacting protein	78/87	WP_030594239 *Streptomyces anulatus*
*acnT1*	346	ABC transporter	87/92	OOQ48072 *Streptomyces antibioticus*
*acnT2*	255	multidrug ABC transporter permease	92/95	OOQ48071 *Streptomyces antibioticus*
*acnT3*	753	UvrABC system protein A	60/77	AMV28079 *Gemmata sp*. SH-PL17
*orf*(+1)	191	GCN5-related *N*-acetyltransferase	81/87	UN35851 *Streptomyces venezuelae*
*orf*(+2)	109	hypothetical protein	74/83	WP_052876109 *Streptomyces sp*. NRRL F-4335
*orf*(+3)	1130	Two-component hybrid sensor and regulator	55/68	SBO98167 *Nonomuraea sp*. ATCC 39727

^a^ Size in units of amino acids (aa); ID/SI: identity/similarity; acn: the BGC of actinomycin from *S. costaricanus* SCSIO ZS0073.

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
