# Peer review of "Identification of the Actinomycin D Biosynthetic Pathway from Marine-Derived Streptomyces costaricanus SCSIO ZS0073"

_marinedrugs, 2019, doi:10.3390/md17040240_

Reviewer 1 Report

This manuscript describes identification and functional analyses of actinomycins biosynthetic gene cluster from a marine derived Streptomyces.

Identification of the biosynthetic gene cluster was performed by whole genome sequencing and homology search based on the known actinomycins biosynthetic genes. Functional analysis of each gene was conducted by systematic knock out experiments using gene disruption method, and HPLC analysis of the extracts of the mutants.

Experimental design was logical, all experiments were performed properly, and every experimental result was very clear. From the obtained experimental results, authors found some essential and non-crucial genes for biosynthesis of actinomycins.

However, to date, many actinomycins biosynthetic gene clusters have been reported from various Streptomyces species, and gene cluster found in this study was quite similar to them. Gene functions clarified in this study were also readily predictable from the previous studies. Large number of mutant strains were prepared in this study, and authors found several non-producing mutants, but high titer or novel analogues producer were not reported.

Considering the above reasons, impact and novelty of this manuscript is not enough to accept as is. Most interesting point of this research is revealing the essentialness of actF and O for the biosynthesis of actinomycins, thus I recommend to re-submit after obtaining some additional information concerning to the function of these genes.

Author Response

Response to Reviewer #1:

Comments and Suggestions for Authors This manuscript describes identification and functional analyses of actinomycins biosynthetic gene cluster from a marine derived Streptomyces. Identification of the biosynthetic gene cluster was performed by whole genome sequencing and homology search based on the known actinomycins biosynthetic genes. Functional analysis of each gene was conducted by systematic knock out experiments using gene disruption method, and HPLC analysis of the extracts of the mutants. Experimental design was logical, all experiments were performed properly, and every experimental result was very clear. From the obtained experimental results, authors found some essential and non-crucial genes for biosynthesis of actinomycins.

Response: We truly appreciate reviewer #1 for giving us such positive comments.

However, to date, many actinomycins biosynthetic gene clusters have been reported from various Streptomyces species, and gene cluster found in this study was quite similar to them. Gene functions clarified in this study were also readily predictable from the previous studies. Large number of mutant strains were prepared in this study, and authors found several non-producing mutants, but high titers or novel analogues producer were not reported. Considering the above reasons, impact and novelty of this manuscript is not enough to accept as is. Most interesting point of this research is revealing the essentialness of actF and O for the biosynthesis of actinomycins, thus I recommend to re-submit after obtaining some additional information concerning to the function of these genes.

Response: We thank reviewer #1 for these suggestions. Actinomycins, commonly employed clinically in anticancer therapeutic regimes, exhibiting excellent antitumor activity. So it has been arousing interest of researchers for years. Although three actinomycin BGCs have been reported, the function of most of the genes were still unknown. Different from those reported, we revealed a new actinomycin BGC from marine Streptomyces and disrupted 24 genes in the BGCs systematically, and interestingly, we elucidated the function of three regulatory genes, acnWU4R, modulating actinomycin production. And the function of acnO and acnF is still to be elucidated in the following research. So, our study is of great importance for improving actinomycin titers and laying the foundation to figuring out how the two 4-MHA pentapeptide forming the actinomycin.

Reviewer 2 Report

Liu and coworkers identified the actinomycin D biosynthetic (acn) gene cluster from marine-derived Streptomyces costaricanus SCSIO ZS0073 using genome sequence analysis and clarified that the identified acn genes are responsible for the actinomycin biosynthesis by the inactivation of several acn genes. Although the identification of the actinomycin biosynthetic gene is not the first, the presently identified gene cluster is different from previous ones and shows additional information. This manuscript is thus publishable in Marine Drugs. There are some minor comments to improve this manuscript before publication. 

1.    AcnM appears to be responsible for the C-methyltransferastion in 4-MHA biosynthesis. A reference “Biochemistry, 2010, 49 (45), pp 9698–9705” regarding this reaction might be referred to propose the function.

2.    Please describe whether a kynurenine 3-monooxygenase homolog or ortholog is encoded in the genome of the producer strain or not.

3.    The function of the acnW and acnU4 are still unclear from the gene inactivation experiments, because the disrupted gene mutants still produce actinomycin D, although the production yield is slightly reduced. A polar effect of the gene expression might be caused. So, maybe the authors should not conclude the functions. 

4.    The HPLC trace of the acnU4 mutant in Figure 3 shows a shoulder peak after compound 1. In addition, there is a small peak around 17 min in the trace. If the authors have some information, please state what those are. 

5.    Similarity, there are small peaks around 17 min and 22 min in the trace of the acnO mutant in Figure 4. If the authors have some information, please state what those are. Because they propose that the acnO and acnF genes are involved in the oxidative dimerization step in the last step of biosynthesis, the accumulated compounds might be biosynthetic intermediates. 

6.    Regarding the oxidative dimerization, is there any possibility whether a certain putative oxidative enzyme that is encoded in the genome of this strain is involved in this step, or nonenzymic oxidation of o-aminophenol like as the reference “PLoS Comput Biol 14(12): e1006672. https://doi. org/10.1371/journal.pcbi.1006672”? It is unclear why monomeric intermediates are not identified in any gene disrupted mutants.

7.    Line 296 in page 10, Frameblot would be FramePlot.

Author Response

Response to Reviewer #2:
Comments and Suggestions for Authors

Liu and coworkers identified the actinomycin D biosynthetic (acn) gene cluster from marine-derived Streptomyces costaricanus SCSIO ZS0073 using genome sequence analysis and clarified that the identified acn genes are responsible for the actinomycin biosynthesis by the inactivation of several acn genes. Although the identification of the actinomycin biosynthetic gene is not the first, the presently identified gene cluster is different from previous ones and shows additional information. This manuscript is thus publishable in Marine Drugs. There are some minor comments to improve this manuscript before publication.

Response: Thanks for your comments. We thank reviewer #2 for the time and efforts you have spent on reviewing our manuscript, this is truly appreciated.

1. AcnM appears to be responsible for the C-methyltransferastion in 4-MHA biosynthesis. A reference “Biochemistry, 2010, 49 (45), pp 9698–9705” regarding this reaction might be referred to propose the function.

Response: Thanks for your suggestion. We have added this reference to the revised manuscript in line 166.

2. Please describe whether a kynurenine 3-monooxygenase homolog or ortholog is encoded in the genome of the producer strain or not.

Response: We appreciate this concern point out by reviewer #2. We analyzed the genome sequence and did not find a kynurenine 3-monooxygenase homolog within the genome.

3. The function of the acnW and acnU4 are still unclear from the gene inactivation experiments, because the disrupted gene mutants still produce actinomycin D, although the production yield is slightly reduced. A polar effect of the gene expression might be caused. So, maybe the authors should not conclude the functions.

Response: We thank reviewer #2 for pointing this out to us. We can judge that the lower production of actinomycin after gene inactivation is not due to polar effect. As we can see from the gene organization of actinomycin BGC, downstream of the acnW locates the boundary gene orf(-1), the disruption of orf(-1) has no effect of actinomycin production. The same as the gene acnU4, downstream of which comes the gene acnU3, which was not necessary for actinomycin biosynthesis. Thus, it is conspicuously not originated from the polar effect that caused the less production of actinomycin D.

4. The HPLC trace of the acnU4 mutant in Figure 3 shows a shoulder peak after compound 1. In addition, there is a small peak around 17 min in the trace. If the authors have some information, please state what those are.

Response: We thank reviewer #2 for pointing this out to us. We noticed the shoulder peak too, and we have analyzed the extract of the mutants by LC-MS, the exact molecular weight of the compound is 1268, consistent with actinomycin X2 that also exist in wild-type strains. In figure 3, we have marked this peak as actinomycin X2.

5. Similarity, there are small peaks around 17 min and 22 min in the trace of the acnO mutant in Figure 4. If the authors have some information, please state what those are. Because they propose that the acnO and acnF genes are involved in the oxidative dimerization step in the last step of biosynthesis, the accumulated compounds might be biosynthetic intermediates.

Response: We thank reviewer #2 for raising this question. We have noticed small peaks around 17 min and 22 min. They are not biosynthetic intermediates. We have knocked out the acnO gene in the fungichromin gene cluster (another main product of S. costaricanus SCSIO ZS0073) in-frame-deletion mutant. Analysis of the mutant extract revealed that the two small peaks are disappeared. Thus, we deduce the small peaks are not intermediates.

6. Regarding the oxidative dimerization, is there any possibility whether a certain putative oxidative enzyme that is encoded in the genome of this strain is involved in this step, or nonenzymic oxidation of o-aminophenol like as the reference “PLoS Comput Biol 14(12): e1006672. https://doi. org/10.1371/journal.pcbi.1006672”? It is unclear why monomeric intermediates are not identified in any gene disrupted mutants.

Response: Reviewer #2 makes an excellent point; we have read the above mentioned article. How the two monomeric intermediates (4-MHA pentapeptide) form the whole structure of actinomycin is certainly unclear. We propose acnO and acnF genes are involved in the oxidative dimerization step in the last step of biosynthesis; these gene products may require the PCP-linked intermediate. Alternatively, the condensation may occur non-enzymatically.

7. Line 296 in page 10, Frameblot would be FramePlot.

Response: Thanks for your suggestion. We have corrected accordingly.

Reviewer 3 Report

In their manuscript entitled "Elucidating the actinomycin D biosynthetic pathway from marine-derived Streptomyces costaricanus SCSIO ZS0073", Liu and colleagues describe the identification of the actinomycin biosynthetic gene cluster and the use of mutant analysis to gain insights into the role of the identified genes on the biosynthesis of the compound. 

The manuscript is well organized, the introduction section has enough detailed information concerning the synthesis of actinomycins. Results are clearly presented and the figures and tables are illustrative of the results. Both the materials and methods section and the conclusions are well presented. The work represents an interesting and well done piece of work, and in my view contribute to a better knowledge on the biosynthesis of actinomycins. There are some issues that need to be fixed prior to publication, which do not reduce the merit of the work. 

line 60: megaterium

line 66: delete  "(a kind of transposon)"

lines 130-132. It is not clear why the acetyl transferase encoded by orf1 is not involved in actinomycin biosynthesis. While it is clearly stated that mutants on orf2 and orf3 are not affected in actinomycin biosynthesis, this is not clear in the case of the orf1 mutant. Do the authors have results showing that the orf1 mutant produce actinomycin at the same level of the wild-type? Please clarify.

line 138:  ...this anthranilic acid derivative...

line 147: Mycobacterium spp.

line 148: acnN1

lines 154-155: Figure legend: info is missing: the scale numbers are kb? The arrows sizes are proportional to gene length? 

line 162: These species? These genes?

lines 162-164: The phrase is confusing, since 4 genes are mentioned, G, H, L, and M, and the sentence states that they are homologous to a set of 5 counterparts...

lines 167-169: In Figure 3, no HPLC record for acnG, acnH, or acnL is presented, only for the acnGHLMP. Either the authors have results clearly indicating that mutants in each gene are unable to produce actinomycin, or the phrase needs to be re-written.

lines 201-203: the sentence is quite confusing. how can the genes act as compensatory elements able to carry out similar functions to those of their products? Please re-write.

figure 3, figure 4 and figure 5 legends: suggestion : HPLC analyses of S. costaricanus strains fermentation extracts.

line 251: Δphs

line 254: is the LbmU fuctioin known? if so, please describe.

line 286: 2000 rpm?

Author Response

Response to Reviewer #3:

Comments and Suggestions for Authors
In their manuscript entitled "Elucidating the actinomycin D biosynthetic pathway from marine-derived Streptomyces costaricanus SCSIO ZS0073", Liu and colleagues describe the identification of the actinomycin biosynthetic gene cluster and the use of mutant analysis to gain insights into the role of the identified genes on the biosynthesis of the compound. The manuscript is well organized, the introduction section has enough detailed information concerning the synthesis of actinomycins. Results are clearly presented and the figures and tables are illustrative of the results. Both the materials and methods section and the conclusions are well presented. The work represents an interesting and well done piece of work, and in my view contribute to a better knowledge on the biosynthesis of actinomycins. There are some issues that need to be fixed prior to publication, which do not reduce the merit of the work.

Response: We appreciate these comments. We thank reviewer #3 for the time and efforts you have spent on reviewing our manuscript; this is truly appreciated.

line 60: megaterium

Response: Thanks for your suggestion. We have corrected accordingly.

line 66: delete "(a kind of transposon)"

Response: Thanks for your suggestion. As reviewer #3 suggested, we have deleted "a kind of transposon".

lines 130-132. It is not clear why the acetyl transferase encoded by orf1 is not involved in actinomycin biosynthesis. While it is clearly stated that mutants on orf2 and orf3 are not affected in actinomycin biosynthesis, this is not clear in the case of the orf1 mutant. Do the authors have results showing that the orf1 mutant produce actinomycin at the same level of the wild-type? Please clarify.

Response: We thank reviewer #3 for pointing this out to us. We have disrupted the orf(+1), orf(+2), orf(+3), and data is in SI, page S16, the mutants of orf(+1), orf(+2), orf(+3) produce almost the same level of actinomycin D as that of in the wild-type strains.

line 138: .this anthranilic acid derivative.

line 147: Mycobacterium spp.

Response: We have corrected above all accordingly.

line 148: acnN1

Response: We thank reviewer #3 for this suggestion. We decided not to revise "acnN1" to "acnN1". The subject of this sentence is NRPS gene, not protein.

lines 154-155: Figure legend: info is missing: the scale numbers are kb? The arrows sizes are proportional to gene length? 

Response: Reviewer #3 makes an excellent point, we have added the ruler line to the picture (Figure 1) in line 154.

line 162: These species? These genes?

lines 162-164: The phrase is confusing, since 4 genes are mentioned, G, H, L, and M, and the sentence states that they are homologous to a set of 5 counterparts.

Response: We thank reviewer #3 for this suggestion, we have corrected above all accordingly.

lines 167-169: In Figure 3, no HPLC record for acnG, acnH, or acnL is presented, only for the acnGHLMP. Either the authors have results clearly indicating that mutants in each gene are unable to produce actinomycin, or the phrase needs to be re-written.

Response: Thanks for your suggestion. We did not disrupted acnG, acnH, and acnL individually. We have rewritten in the main text to read "Inactivation of acnGHLMP in combination, followed by metabolite analysis of the mutant strain revealed ...)".

lines 201-203: the sentence is quite confusing. how can the genes act as compensatory elements able to carry out similar functions to those of their products? Please re-write.

Response: We thank reviewer #3 for raising this question. According to the inactivation results of three transporter genes, the yields of actinomycin D is almost the same as that of in S. costaricanus SCSIO ZS0073 wide type. It seems that these genes have no function to export the compound to the extracellular. We have rewritten the sentence to read as "Consequently, we concluded that these three transporter genes are not necessary in actinomycin D biosynthesis".

figure 3, figure 4 and figure 5 legends: suggestion: HPLC analyses of S. costaricanus strains fermentation extracts.

Response: Thanks for your suggestion. We have corrected accordingly.

line 251: Δphs

Response: Thanks for your suggestion. We have corrected accordingly.

line 254: is the LbmU fuctioin known? if so, please describe.

Response: We appreciate this input from reviewer #3. We have added these sentences to read "LmbU is a regulatory gene involved in licomycin biosynthesis in S. lincolnensis 78-11. It always contains a TTA codon close to the N-terminal end of its ORF. The codon is often found in genes involved in the regulation of differentiation or secondary metabolism [42]" in the main text.

line 286: 2000 rpm?

Response: We have revised "2000 rpm" to "200 rpm" throughout complete manuscript.

Round  2

Reviewer 1 Report

It is true that actinomycin D is an important anti-tumor drug, and development of the high-titer producing strain is desired. It was appreciated that the all genes responsible for the biosynthesis of actinomycins were clearly confirmed by systematic gene disruption analysis.

However, these experiments did not reveal any molecular mechanisms of enzymes or protein functions. For example, yield of actinomycins decreased in acnW, U4, and R mutants, but it was not clear that this change was result of the knock-out of the positive regulators, or just a disruption of the genomic structure. Authors need to do function recovery experiments by transforming corresponding genes to the knock-out strains, or construct over-expression strains of each gene to confirm they are actually act as positive regulators.

Additionally, authors mentioned that acnF and O are supposed to be involved in the dimerization of the 4-MHA-pentapeptide monomers, and it is very interesting idea. But, any experimental data was not shown to support this hypothesis. Moreover, there was not a major peak in the HPLC chromatograms in Figure 4x and 4xi, suggesting that no corresponding monomer produced. In order to confirm the accumulation of the monomer, authors should analyze the extracts of all mutants by LC-MS.

Author Response

Comments and Suggestions for Authors:

   It is true that actinomycin D is an important anti-tumor drug, and development of the high-titer producing strain is desired. It was appreciated that the all genes responsible for the biosynthesis of actinomycins were clearly confirmed by systematic gene disruption analysis. However, these experiments did not reveal any molecular mechanisms of enzymes or protein functions. For example, yield of actinomycins decreased in acnW, U4, and R mutants, but it was not clear that this change was result of the knock-out of the positive regulators, or just a disruption of the genomic structure. Authors need to do function recovery experiments by transforming corresponding genes to the knock-out strains, or construct over-expression strains of each gene to confirm they are actually act as positive regulators.

Response: We thank reviewer #1 for the time and efforts that have spent on reviewing our manuscript.

    Although two actinomycin D gene cluster have been reported by Ullrich Keller from S. antibioticus IMRU 3720 and S. chrysomallus (J. Bacteriol. 2010, 192, 2583-2595; Adv. Appl. Bioinform. Chem. 2017, 10, 29), the function of each individual gene has never been studied in vivo through gene inactivation experiments.

    In our paper, we report a third pathway for actinomycin D biosynthesis from S. costaricanus SCSIO ZS0073. For the first time, we systematically disrupted 25 genes individually in the actinomycin D biosynthetic gene cluster. Metabolomics analysis of each mutant fermentation extract led us to identify: i) six boundary genes orf (-3), orf (-2), orf (-1), orf(+1), orf(+2), and orf(+3); ii) two NRPS genes acnN1 and acnN3, iii) one cytochrome P450 encoding gene acnP; iv) seven genes that are not necessary for actinomycin biosynthesis including acnA, B, U1, U2, U3, Q, and phs; v) three transportion genes acn T1, T2, T3, and vi) three positive regulatory genes acnW, U4, R. These represent tremendous amount of work and significant contributions to the in depth study of the actinomycin biosynthetic mechanism; whether these genes are involved or not in the actinomycin D pathway have been largely unknown previously.

    As we can see from the gene organization of actinomycin gene cluster, downstream of the acnW locates the boundary gene orf(-1), the disruption of orf(-1) has no effect on actinomycin production. The same as the gene acnU4, downstream of which comes the gene acnU3, which was not necessary for actinomycin biosynthesis. Thus, it is conspicuously not originated from the disruption of the genomic structure that caused the decreased actinomycin production. Hence, in the manuscript we wrote “AcnR encodes a TetR family transpcriptional regulator and usually acts as a transcriptional repressor or activator in many biological processes such as cell-cell communication and metabolite regulation; ....."On the basis of these findings, AcnW and AcnU4 appear to function as positive regulators of actinomcyin D production although the structural families to which they belong are not yet apparent.”

    As for function recovery experiments, we found the natural actinomycin producer S. costaricanus SCSIO ZS0073 is recalcitrant to complementary experiments. We had ever performed complementary experiments for acnR; the result revealed that exogenous plasmid carrying target gene could not be introduced into the genome of S. costaricanus SCSIO ZS0073.

    To more accurately reflect the content of our discovery, we decide to revise the title of our paper to "Identification of the Actinomycin D Biosynthetic Pathway from Marine-derived Streptomyces costaricanus SCSIO ZS0073".

Additionally, authors mentioned that acnF and O are supposed to be involved in the dimerization of the 4-MHA-pentapeptide monomers, and it is very interesting idea. But, any experimental data was not shown to support this hypothesis. Moreover, there was not a major peak in the HPLC chromatograms in Figure 4x and 4xi, suggesting that no corresponding monomer produced. In order to confirm the accumulation of the monomer, authors should analyze the extracts of all mutants by LC-MS.

Based on our systematic gene inactivation experiments, we conclude in the paper that "AcnO and AcnF are genes essential to actinomycin D biosynthesis but whose functions are unknown". The mutants of these two genes fully abolished the actinomycin D and analogs production. Moreover, we have analyzed the extracts of DacnO and DacnF mutants by LC-MS previously, and we did not found the accumulation of the monomer. Thus, we propose these two genes may catalyze the cyclization, dimerization and oxidation steps using the the PCP-tethered intermediate as substrate; the terminal TE domain, acnO, and AcnF might be involved in these transformation steps. We have revised figure 2 in the manuscript in page 4 to support our experimental results. In the paper, we wrote "We propose these two genes might be involved in the dimerization of the 4-MHA-pentapeptide monomer en route to actinomycin (Figure 2). Efforts to identify the exact roles played by these gene products are currently ongoing and will be reported in due course".

Revised figure 2.

Round  3

Reviewer 1 Report

So far, several actinomycin biosynthetic gene clusters have been reported from various Streptomyces sp., but systematic gene knock-out experiment is unprecedented. It should be evaluated to find an essential but function unknown gene, acnF, by detailed gene disruption analysis.

General biosynthetic pathway of actinomycins till formation of the pentapeptide lactone monomer has been already revealed and readily predictable from the genetic information, but the last step, dimerization and formation of a phenoxadinone chromophore, is largely unknown in spite of the intensive studies. So, authors should be careful and provide reasonable evidences to mention about the mechanisms of this step.

Considering the provided experimental results, acnF might be involved in this last step, but no positive results were obtained yet. Authors proposed dimerization would occur before cleavage from the PCP domain, because no free-monomer was detected by LCMS analysis. In this case, acid-hydrolysis of the crude protein extract would work to confirm this hypothesis. 

Considering the reasons mentioned above, after revision of the following points, the manuscript could be acceptable.

1. AcnO is categorized into function unknown group, but judging from the homology information, it is most likely regulator. The reason is not clear enough why authors categorized it into the function unknown group.

2. LCMS data of the acnF knock-out strain should be provided to explain the proposed biosynthetic pathway shown in Figure 2. And if possible, LCMS result of the acid- hydrolysate of the crude protein extract should be shown.

3. Is acnF conserved in other actinomycin biosynthetic gene clusters so far reported?

4. In figure 1, acnF and P are colored as 4-MHA biosynthetic genes.

Author Response

Comments and Suggestions for Authors:

So far, several actinomycin biosynthetic gene clusters have been reported from various Streptomyces sp., but systematic gene knock-out experiment is unprecedented. It should be evaluated to find an essential but function unknown gene, acnF, by detailed gene disruption analysis.

General biosynthetic pathway of actinomycins till formation of the pentapeptide lactone monomer has been already revealed and readily predictable from the genetic information, but the last step, dimerization and formation of a phenoxadinone chromophore, is largely unknown in spite of the intensive studies. So, authors should be careful and provide reasonable evidences to mention about the mechanisms of this step.

Considering the provided experimental results, acnF might be involved in this last step, but no positive results were obtained yet. Authors proposed dimerization would occur before cleavage from the PCP domain, because no free-monomer was detected by LCMS analysis. In this case, acid-hydrolysis of the crude protein extract would work to confirm this hypothesis. 

Considering the reasons mentioned above, after revision of the following points, the manuscript could be acceptable.

Response: We appreciate these comments.

1. AcnO is categorized into function unknown group, but judging from the homology information, it is most likely regulator. The reason is not clear enough why authors categorized it into the function unknown group.

Response: We appreciate this suggestion. We have categorized the acnO into the “regulatory and self-resistance genes” in the revised manuscript, and figures 1, 2, 3, 4 are adjusted accordingly.

 2. LCMS data of the acnF knock-out strain should be provided to explain the proposed biosynthetic pathway shown in Figure 2. And if possible, LCMS result of the acid- hydrolysate of the crude protein extract should be shown.

Response: We thank reviewer #1 for this suggestion. We have added the LCMS data of the acnF knock-out strain as Figure S29 in the Supplementary material in page 27. The molecular weight of the pentapeptide lactone monomer is 630.3. Thus, we extracted the masses of [M+H]+ (631.3) in the ESI-MS data of the fermentation extract; the result is shown in panel c.

Figure S29 HPLC-ESIMS chromatogram of the fermentation extract of ∆acnF. a) HPLC profile of the extract of ∆acnF (λ = 254 nm); b) total ion chromatogram (TIC) of the extract of ∆acnF; c) extract ion chromatogram (EIC) of the pentapeptide lactone monomer with [M + H]+ at m/z 631.3.

3. Is acnF conserved in other actinomycin biosynthetic gene clusters so far reported?

Response: We thank reviewer #1 for pointing this out to us. According to the reference, acnF is conserved in other actinomycin biosynthetic gene clusters. Its counterparts are saacmT in S. antibioticus, scacmT in S. chrysomallus, and AcmG7’ in Streptomyces iakyrus.

4. In figure 1, acnF and P are colored as 4-MHA biosynthetic genes.

Response: We thank reviewer #1 for pointing this out to us. We have changed the colors of acnF and acnP in figure 1 in revised manuscript.

We would like to thank you and the referee for critical reading and comments of the manuscript. We hope the revised manuscript has met your expectations.